# Implementing Core Genes and an Omnigenic Model for Behaviour Traits Prediction in Genomics

**DOI:** 10.3390/genes14081630

**Published:** 2023-08-16

**Authors:** Tautvydas Rancelis, Ingrida Domarkiene, Laima Ambrozaityte, Algirdas Utkus

**Affiliations:** Institute of Biomedical Sciences, Faculty of Medicine, Vilnius University, Santariskiu Str. 2, LT-08661 Vilnius, Lithuania; ingrida.domarkiene@mf.vu.lt (I.D.); laima.ambrozaityte@mf.vu.lt (L.A.); algirdas.utkus@mf.vu.lt (A.U.)

**Keywords:** behaviour traits, core genes, omnigenic model, complex inheritance, biological pathways, hormones, neurotransmitters, enzymes

## Abstract

A high number of genome variants are associated with complex traits, mainly due to genome-wide association studies (GWAS). Using polygenic risk scores (PRSs) is a widely accepted method for calculating an individual’s complex trait prognosis using such data. Unlike monogenic traits, the practical implementation of complex traits by applying this method still falls behind. Calculating PRSs from all GWAS data has limited practical usability in behaviour traits due to statistical noise and the small effect size from a high number of genome variants involved. From a behaviour traits perspective, complex traits are explored using the concept of core genes from an omnigenic model, aiming to employ a simplified calculation version. Simplification may reduce the accuracy compared to a complete PRS encompassing all trait-associated variants. Integrating genome data with datasets from various disciplines, such as IT and psychology, could lead to better complex trait prediction. This review elucidates the significance of clear biological pathways in understanding behaviour traits. Specifically, it highlights the essential role of genes related to hormones, enzymes, and neurotransmitters as robust core genes in shaping these traits. Significant variations in core genes are prominently observed in behaviour traits such as stress response, impulsivity, and substance use.

## 1. Introduction

Understanding complex behaviour traits is a very important part of human genomics research. Behaviour traits, such as impulsivity, stress, depression, and addictive tendencies, profoundly affect individual lives and social interactions. However, their multifactorial nature, influenced by both genetic and environmental factors and the vast number of genetic variants involved, presents significant challenges in elucidating their genetic underpinnings.

This review is grounded in a narrative approach, and it explores the emerging concept of core genes and the omnigenic model, aiming to shed light on the genetic basis of behavioural traits and providing an original perspective on the genetic architecture of complex traits, moving beyond the traditional genome-wide association studies (GWAS) and polygenic risk score calculations that have dominated the field. The interplay of core genes within intricate biological pathways is investigated, with the goal of achieving a deeper understanding of behaviour characteristics. This approach, focusing on genes with clear biological pathways and significant influence on behaviour, suggests a more standardised approach for the practical prediction of behaviour traits.

In the following sections, the review discusses the intricacies of genome-wide association studies, the practical application of combined genome variants for complex traits, and the implementation of the omnigenic model. The review also presents the significance of biological pathways and the role of core genes in shaping behaviour traits. It is intended to provide a comprehensive understanding of the current state of attempts to predict behavioural traits using genomics and the potential future directions in this field.

## 2. Complex and Monogenic Traits

Unlike a monogenic trait, complex traits are not explained by one or a few genes but are characterised by polygenicity, where multiple genetic variants contribute to the trait’s manifestation. Furthermore, studying complex traits is challenging because it is determined by the combination of genetic and environmental factors [1].

The genomic variation in monogenic traits is widely used for practical purposes such as genetic counselling, diagnostic testing, carrier screening, or personalised medicine. The availability of such wide practical use comes due to the simplistic nature of monogenic trait manifestation and the constantly improving quality of information collected by researchers, which is stored in public databases such as ClinVar [2].

## 3. The Role and Limitations of Genome-Wide Association Studies

The field of complex inheritance genomics has experienced a substantial surge in available data. A key contributing factor for associating thousands of genome variants with complex traits has been the widespread implementation of genome-wide association studies (GWAS) [3,4].

The usual way in which GWAS data are utilised on an individual level is by calculating the polygenic risk score (PRS). The PRS is a quantitative measure that combines the effects of multiple genetic variants across the genome to estimate an individual’s genetic predisposition to a particular trait. Researchers assign weights to each genetic variant based on their association with the trait from the GWAS study, sum up the weighted effects of all the genomic variants related to that trait, and the resulting single numeric value represents the individual’s possible predisposition to the trait [5,6].

However, research has shown that in many cases of GWAS, it is challenging to replicate results and genomic data retrieved from GWAS could have contradictory results. Most importantly, having thousands of variants for one trait complicates the phenotypic prediction from such genome data.

The genetics of physical performance is an excellent example of too many associated variants from GWAS data. Bray and colleagues reviewed numerous studies from the year 2007 and compiled a list of 214 genetic markers associated with physical performance [7]. However, Varillas-Delgado et al. highlighted the limited reproducibility of reported associations and many “false positive” findings. Similar conclusions were raised in a publication by Rankinen et al. in the *Journal of Physiology*, where the authors noted that while the number of genes and polymorphisms associated with physical performance continues to grow, the predictive power of these genetic markers is limited, and it is problematic to translate genetic research from GWAS into practical use [8,9]. A 2023 study by Semenova and colleagues reported that out of 251 DNA polymorphisms related to athlete status, only 128 markers consistently showed positive associations in at least two separate studies. Moreover, they provided a timeline of athletic-performance-associated genome variants, highlighting that numerous variants previously linked to physical performance were not included in the 2023 list [10].

Due to this, commercial genetic companies in their sports genetic testing tend to focus only on the main known genes, such as *ACTN3* and *ACE*, which have very clear biological pathways related to physical performance, instead of calculating polygenic risk scores from associated variants from many genes [11].

## 4. Practical Application of Combined Genome Variants for Complex Traits

One of the better practical examples of utilising a combined genomic variation of complex traits for practical purposes is the forensic DNA Snapshot™ developed by Parabon NanoLab [12]. The DNA Snapshot™ system, which received funding from the US Department of Defense, uses machine learning algorithms to analyse complex trait-associated genetic variation. This advanced technology enables the system to predict various physical features, including genetic ancestry; eye, hair, or skin colour; freckling; and face shape. Combining these predictions, the system generates a composite profile known as a “genetic photo robot,” estimating an individual’s physical characteristics. While it is just a prediction tool to predict the physical appearance of complex traits from individual DNA data, the prediction is accurate and has direct practical use in forensics [12,13].

Just as the DNA Snapshot™ system predicts physical appearance traits from DNA data, a similar approach could be taken to predict behaviour traits. By analysing the genomic variation associated with behaviour characteristics, such as impulsivity, risk taking, or addictive tendencies, this tool could provide insights into an individual’s behaviour profile.

Understanding human behaviour involves a complex interplay of genetic and environmental factors. Genetic information can offer valuable insights into a person’s behaviour predispositions and vulnerabilities, providing a tool, e.g., for awareness and self-education to improve personal well-being. By being aware of the genetic component of their behaviour, individuals can control and avoid environmental triggers associated with those specific behaviour traits [14]. This knowledge empowers individuals to make informed choices and encourages them to choose environmental aspects, ultimately helping them manage and shape their behaviour more effectively.

However, complex behaviour traits pose even more significant challenges compared to physical appearance traits due to the more decisive influence of environmental factors and the involvement of a higher number of genetic variants. The conventional approach of calculating PRSs from all available GWAS data has limited practical usability in the case of behaviour traits due to statistical noise and the small effect size from a high number of genome variants involved.

## 5. Omnigenic Model Implementation

Creating a personal behaviour profile from GWAS data is very challenging. An omnigenic model suggested in 2017 by Boyle and his colleagues could be applied to predicting complex traits from genomic data, which expands the view of complex traits from polygenic to omnigenic [15].

The omnigenic model attempts to explain this observation by suggesting that diseases can be thought of as networks, where genes directly involved in complex traits are named “core genes”, while peripheral genes are spread through whole-genome interaction networks. The omnigenic model represents a paradigm shift in our understanding of the genetic basis of complex traits.

Some researchers criticised this model as although it looks appealing, it oversimplifies complex traits, underestimates their biological complexity, and should not be focused on discovering only core genes [16].

Despite receiving some criticism, the omnigenic model has demonstrated its usefulness in various studies. For instance, Ratnakumar and his colleagues utilised the omnigenic model and successfully identified novel disease-associated core genes [17]. Studies of *Populus nigra* and *Drosophila melanogaster* support the omnigenic model [18,19]. The omnigenic model contributed to suggesting causes for disorders such as schizophrenia and autism [20,21].

From the perspective of complex traits in behaviour genetics, narrowing the focus solely to core genes and disregarding peripheral genes with very small effect sizes may reduce the accuracy compared to a full polygenic risk score encompassing all trait-associated variants. However, this simplification enables a more straightforward and easier practical implementation.

## 6. Biological Pathways for Behaviour Traits

When discussing core genes in the context of behaviour, we are referring to genes that possess well-defined biological pathways and play significant roles in critical processes associated with behaviour. These genes substantially influence behaviour by modulating various intricate biological mechanisms and pathways.

Several biological processes contribute to the complex interplay of factors influencing behaviour, such as neurotransmission, synaptic plasticity, hormone regulation, and neuronal signalling [22,23].

To elaborate further, it is important to underscore the roles of particular biochemical elements, which are fundamental to these processes. Enzymes, hormones, and neurotransmitters are essential components within an organism’s chemical communication and regulation systems, constituting vital contributors to the manifestation of behaviour traits. An enzyme is a protein that acts as a catalyst, facilitating and speeding up biochemical reactions in the body. It plays crucial roles in various metabolic processes, helping to break down substances, build new molecules, and regulate cellular activities [24]. A hormone is a biological compound that serves as a regulatory messenger in multicellular organisms, organising, coordinating, and controlling cellular and tissue functions. These chemical messengers are crucial in various physiological processes, including metabolism and behaviour [25]. While hormones may modulate the expression of behaviour, they are not the cause of behaviour itself. Different behaviours are driven by a variety of internal and environmental stimuli, with hormones playing a prominent role in regulating and influencing behaviour responses [26].

A neurotransmitter is a chemical messenger that is synthesised and released by neurons and is used in the process of synaptic communication between neurons. It is involved in the process of sensory information and motor behaviour control [27].

While hormones and neurotransmitters share the role of chemical messengers, hormones exert systemic effects by being released into the bloodstream, whereas neurotransmitters act locally within the nervous system to transmit signals between neurons. Nevertheless, the distinction between hormones and neurotransmitters can be ambiguous, as certain substances like epinephrine and dopamine can serve dual roles as hormones and neurotransmitters [28].

## 7. Core Genes of Behaviour

A good example of core behaviour genes with clear biological pathways related to enzymes can be found in genes associated with substance abuse. The aldehyde dehydrogenase (ALDH) enzyme is involved in the metabolism of alcohol, and gene variations related to it have been found to influence the risk of alcohol dependence [29].

Risk of alcohol dependence, while being a complex trait, can be influenced by only two known genome variants that significantly impact its risk. Such variants are also interesting due to their distinct frequencies in East Asia compared to other populations worldwide, leading to a lower risk of alcohol dependence in East Asia [30,31].

rs1229984 is a single-nucleotide polymorphism (SNP) in the *ADH1B* gene, which encodes a subunit of the alcohol dehydrogenase enzyme, a crucial catalyst in hepatic ethanol metabolism. The T allele of this SNP enhances the enzyme’s activity, accelerating the metabolism of alcohol to acetaldehyde, a toxic by-product. Elevated acetaldehyde levels may cause symptoms such as facial flushing, nausea, and elevated heart rate, which deter individuals from heavy drinking and thus lower the risk of developing alcohol dependence. Similarly, rs671 is an SNP in the *ALDH2* gene, known for its impact on the alcohol metabolism process. The A allele leads to a Glu504Lys substitution that significantly impairs enzyme activity. This results in an accumulation of acetaldehyde when alcohol is consumed, provoking symptoms that deter heavy drinking and may lower the risk of alcohol dependence [29,30,32]. In the genome aggregation database (gnomAD), the frequency of the rs1229984 T allele in the East Asia (A) population is 73.9%, compared with only 3.8% in the European population, and the rs671 A allele is 25.4% compared with 0.003% [33].

Figure 1 represents the interplay of genetic and environmental factors in alcohol dependence. The manifestation of complex behaviour traits is influenced by the interaction between core and peripheral genes, with variations in core genes having a significantly higher impact. Specific alleles of rs1229984 and rs671 in the *ADH1B* and *ALDH2* genes, accordingly, lead to physical reactions like facial flushing and nausea after alcohol consumption, lowering the risk of alcohol dependence. In contrast, alternative alleles that do not cause these physical discomforts increase the likelihood of individuals becoming alcohol-dependent, as they lack these immediate negative physical deterrents post alcohol consumption. Environmental factors, which can be modulated, significantly influence alcohol consumption and dependence, particularly in groups with a higher genetic predisposition to alcohol dependence. These factors can either promote alcohol consumption, thereby increasing dependence risk, or discourage it, consequently reducing the risk of alcohol dependence.

Prominent among the robust core genes associated with behaviour are those involved in hormone regulation. A compelling illustration of such genes is those related to stress. The *FKBP5* gene is primarily associated with regulating the stress response and is important in releasing stress hormones, including cortisol. Its variants are highly associated with stress response, for example, the rs1360780 variant in which the G allele has been linked with impaired regulation of the stress hormone cortisol and, due to this, has been associated with increased vulnerability to stress-related disorders [34,35]. The *CRHR1* gene, which encodes the corticotropin-releasing hormone receptor 1, is associated with increased reaction to stress. A good example of this gene is the T allele of rs110402, which causes increased sensitivity to stress hormone signalling [34,36]. The *NR3C1* gene encodes the glucocorticoid receptor, which is involved in binding and responding to glucocorticoid hormones. Its variant, rs6195, plays a role in regulating and responding to cortisol. The C allele of rs6195 has been associated with increased stress sensitivity and a higher risk for stress-related disorders [34,37].

Many genes are intricately linked to their functioning within the neurotransmitters domain, making them highly relevant as core determinants of behaviour traits. The *CHRNA5*/*CHRNA3*/*CHRNB4* gene cluster could represent the core genes for nicotine abuse and addiction. These genes encode the nicotinic acetylcholine receptor subunits, which are involved in synaptic neurotransmission. Genome variants such as rs16969968 in *CHRNA5* have been strongly associated with increased nicotine dependence and with a higher risk of developing a nicotine addiction. As well as rs578776, rs1051730 variants in the *CHRNA3* gene are likewise associated with developing nicotine addiction [38].

Another example of core genes related to neurotransmitters is those related to impulsivity. The *COMT* gene is involved in the metabolism of catecholamine neurotransmitters, such as dopamine, epinephrine, and norepinephrine. Its variant rs4680 is known to be associated with impulsive behaviour [39,40]. The rs25531 variant of the *SLC6A4* gene, which encodes the serotonin transporter protein, is also associated with impulsivity [41]. This particular gene, and even the same rs25531 variant, is also associated with other behaviour traits such as anxiety, depression, and suicide [42,43]. These and other examples of the core genes and their variants associated with different behaviour traits can be found in Table 1. The behaviour traits included in Table 1 were selected based on their relevance to the topic of this review and the robustness of the scientific evidence linking them to specific core genes and genomic variants.

## 8. Core Genes: Ethnic and Gender Perspectives

The primary emphasis of this review is to simplify and standardise the prediction of complex behavioural traits considering genetic factors. To achieve this, priority is given to the variation observed in the core genes. The core genes are characterised by their direct biological pathways and the coding genes responsible for producing specific products. Any variation within these genes directly influences the associated biological processes and is likely to have a universal effect across all individuals [66]. By focusing solely on the core genes, we can minimise the scope of factors under consideration, thereby streamlining our analysis and creating a more uniform approach to predicting behaviour traits from genetic factors. However, even in the case of the core genes, it is important to consider additional factors, such as gender or ethnicity, which are considered to have a high impact on behaviour manifestation.

Considering the gender factor, gender-based genetic variation is a notable aspect. Sex chromosomes (X and Y) hold some genes that are not present in the opposite gender, leading to sex-specific genetic effects. For instance, the monoamine oxidase A (*MAOA*) gene located on the X chromosome is known to influence behavioural traits such as aggression and impulsivity, impacting males and females differently. Namely, greater risky behaviour is found in males [67]. Gender differences in hormone levels can also affect the expression and functioning of core genes, thus affecting behaviour. For instance, the stress response, which is influenced by variations in the *FKBP5*, *CRHR1*, and *NR3C1* genes, can vary between males and females due to the differences in the regulation of cortisol, a hormone that plays a key role in the stress response [68,69]. Variants of the *COMT* gene, involved in the metabolism of catecholamine neurotransmitters, have been implicated in impulsivity. However, it has been suggested that the impact of *COMT* variants on impulsivity may differ between genders, warranting further investigation into gender-specific effects [70]. However, it is important to note that gender differences in trait manifestation do not change the underlying role of the core genes involved in particular traits.

In terms of ethnicity, given that core genes have a direct influence on corresponding biological processes, the effect of a specific genomic variation within these genes tends to be universal, irrespective of an individual’s ethnic group. While the impact of a specific variant remains constant across various ethnic groups, the frequency of these variants can differ significantly among different populations. Genetic diversity and unique evolutionary histories among different populations can contribute to substantial variation in the frequency of these variants without altering the effect of the specific variant. A clear example of this is the distinct frequencies of previously described variants associated with alcohol dependence, such as rs1229984 in the *ADH1B* gene and rs671 in the *ALDH2* gene, observed in East Asia compared to other populations worldwide [30,31].

## 9. Conclusions

The omnigenic model and the selection of core genes with clear biological pathways present a promising approach to studying complex traits. Integrating core genes with other behaviour datasets presents a more precise approach that avoids genome variation with very small effect sizes. This approach could help to reduce statistical noise and the wide range of statistical methodologies used, thereby paving the way for the standardisation of complex trait analysis.

However, it is crucial to note that numerous studies have demonstrated that the majority of the genetic variation associated with complex traits is located in non-coding DNA regions and genes, which are not considered core genes. This may be attributed to the fact that the associated variants are proximal to core genes, and genes have a direct pathway with the main core genes [71,72,73]. For example, one of the strongest associations between impulsivity and genetic variants is in the *CADM2* gene, which mediates synaptic plasticity. The gene is co-expressed with *HTR2A* and *GABRA2*, both of which are also implicated in impulsivity. All three genes are involved in neurological processes, with *HTR2A* and *GABRA2* playing direct roles in neurotransmitter signalling pathways. Specifically, *HTR2A* is a key component in the serotonin signalling pathway, while *GABRA2* is integral to the γ-aminobutyric acid (GABA) signalling pathway. This co-expression and shared involvement in neurotransmission suggest potential synergistic roles in the modulation of impulsive behaviours [74,75,76].

Direct pathogenic variants in core genes predisposing complex traits are relatively rare, as important protein-coding sequences tend to be conservative. However, when such variants occur in these genes, they can have a significant effect on trait manifestation. For instance, the hormone vasopressin gene *AVPR1A* variants are associated with autism spectrum disorder [77]. In such cases, identifying core genes and genetic variants can contribute to understanding different conditions and disorders as well as guide the development of personalised interventions and treatments. Because of the drastic interference in traits, the majority of the genetic variation associated with particular traits is not found directly within the core gene but rather in genes (and their variants) that are interlinked with it. Publications on the omnigenic model have demonstrated that variants identified in GWAS studies with the highest *p* values tend to be located in the proximity of the core gene rather than directly within it. This suggests the presence of intricate genetic networks and interactions surrounding the core gene, contributing to the complexity of trait manifestation [66].

Looking forward, it is worth considering the possibility that the focus should shift from genome-wide association studies (GWAS) and genome-wide genotyping techniques towards studies of the whole exome or genome. This could provide a more comprehensive understanding of the genetic underpinnings of complex traits.

## Figures and Tables

**Figure 1 genes-14-01630-f001:**
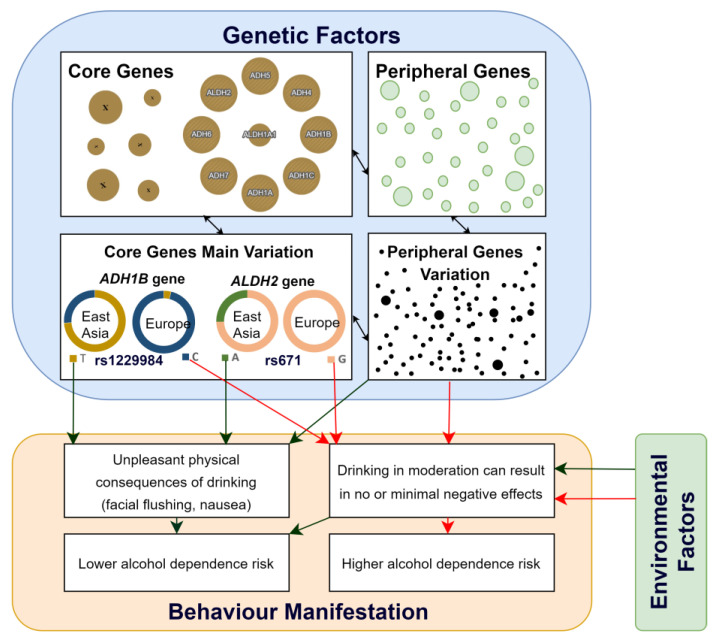
The diagram illustrates the interplay of genetic and environmental factors in alcohol dependence. “Core Genes” represent the primary genes associated with alcohol consumption, while “Core Genes Main Variation” displays the main genome variants of *ADH1B* and *ALDH2* genes and their allele distribution in different populations. “Peripheral Genes” and “Peripheral Genes Variation” indicate the multitude of genes that influence this complex trait, albeit with a lesser impact. Black arrows indicate interactions between different elements or factors. The green arrows point to factors associated with a decreased risk of alcohol dependence. The red arrows highlight factors that increase the risk of alcohol dependence.

**Table 1 genes-14-01630-t001:** Core genes and variants associated with behaviour traits.

Behaviour Trait	Core Genes	Genome Variants	References
Alcohol Use	*ADH1B*, *ADH1C*, *ALDH2*	rs1229984, rs698, rs671	[30,31,44]
Smoking	*CHRNA5*, *CHRNA3*, *CHRNB4*	rs16969968, rs1051730	[38,45,46,47]
Drug Use	*DRD2*, *OPRM1*, *ABCB1*	rs1800497, rs1799971, rs1045642	[48,49,50]
Stress	*FKBP5*, *CRHR1*, *NR3C1*	rs1360780, rs110402, rs6195	[51,52,53]
Anxiety	*SLC6A4*, *HTR1A*, *HTR2A*	rs6295, rs6311, rs6313	[54,55,56]
Fearfulness	*SLC6A4*, *COMT*, *MAOA*	rs4680, rs6323, rs6354	[57,58,59]
Impulsivity	*DRD4*, *SLC6A4*, *COMT*, *HTR2A*	rs1800955, rs25531, rs4680	[40,41,60]
Lack of Sleep	*ABCC9*, *PER2*, *PER3*, *CRY1*	rs11046209, rs934945, rs228697	[61,62,63]
Depression	*SLC6A4*, *BDNF*, *HTR2A*	rs25531, rs6265, rs6313	[42,64,65]

## Data Availability

Not applicable.

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
