# Peer review of "Implementing Core Genes and an Omnigenic Model for Behaviour Traits Prediction in Genomics"

_genes, 2023, doi:10.3390/genes14081630_

Round 1

Reviewer 1 Report

Major Comments:

1.      The authors should clarify the importance of the review in the ‘Introduction’ section.

2.      The behavioral traits are not clearly mentioned in the ‘Introduction’!

3.      Are there any variations in the core genes and variants associated with behavioral traits between male and female? If so, please mention in the Table 1 and discuss about the potential causes.

4.      Are core genes and variants associated with behavioral traits are similar across the ethnic groups? Please discuss.

5.      How did the authors select behaviors mentioned in the Table 1? Is there any specific reason of selection these behaviors?

6.      How these methods will help personalized disease conditions?

Major Comments:

1.      The authors should clarify the importance of the review in the ‘Introduction’ section.

2.      The behavioral traits are not clearly mentioned in the ‘Introduction’!

3.      Are there any variations in the core genes and variants associated with behavioral traits between male and female? If so, please mention in the Table 1 and discuss about the potential causes.

4.      Are core genes and variants associated with behavioral traits are similar across the ethnic groups? Please discuss.

5.      How did the authors select behaviors mentioned in the Table 1? Is there any specific reason of selection these behaviors?

6.      How these methods will help personalized disease conditions?

Reviewer 2 Report

The core genes and omnigenic model has been recently topics. It seemed to be nice to describe about behaviors based on the recent model.

1.      This was just a narrative review. The choice of thesis (behaviors) and references should be systematically explained.

2.      This reviewer wondered if summarizing the thesis as behaviors is news in the omnigenic model. If so, the authors may more emphasize it as the news.

3.      The way the literature was cited might be unusual. The literature was often introduced in the last sentence of the paragraph. Or there appeared to be no references where the (non-expert) readers could need; for instance, page 2, row 56 (after …the associated genes) and row 58 (after …positive findings), row 70 (after …NanoLab), row 79 (after …traits), page 4, row 156 (after…dependence; this paragraph did not have any references) etc.

4.      Was Table 1 original by the authors (or not)? If it was original, how were the contents created (for example, via meta-analysis). Furthermore, the authors could more state the original comments regarding expectation to the use of the omnigenic model (compared with the conventional model) with reference to the Table 1.

5.      Page 2 row 51; the authors mistyped ‘Gene’ (change the G to g). Check mistypes throughout the paper.

6.      When using the omnigenic model, how can the intervention of behaviors change in the real word? The authors might state more perspectives if possible.

English can be recheck throughout the paper. Rather, editing the paper style (including the reference citation) might be reconsidered.

Reviewer 3 Report

This review is very interesting and complete.

I suggested to  clarify the aim of this paper in a short paragraph. Furthermore a section about methods is missing. Please add a paragraph about the choice of papers used and about the bibliography research.

Insert some more recent paper (2/3 are ok)

It is quite ok

Round 2

Reviewer 1 Report

The manuscript has been improved and can be accepted for publication.

The manuscript has been improved and can be accepted for publication.

Author Response

Thank you for your review and feedback.

We're pleased to hear the manuscript is suitable for publication and appreciate your guidance in this process.

Reviewer 2 Report

The paper has been improved. The authors should more state that their review paper was based on the narrative and authorized review. The drawback (i.e., in general, about neutrality of evidence and fact, potential selection bias of literature and thematic items – selected behaviors items in cases) can be more added in the text.

Author Response

Thank you for your careful review and insightful comments on our manuscript. We value your perspective and appreciate the depth of your feedback.

We accept the concern of the reviewer of the narrative approach in the manuscript and added a sentence in the manuscript to emphasise it as well. While we fully agree that traditional GWAS and polygenic risk score calculation methods are valuable methods that have significantly advanced genetics and complex trait analyses. These methods are well-elucidated in the literature and have become standard practice. In our article, we want to focus on a discussion about behaviour traits analysis from an omnigenetic perspective. While simplifying genome variation may reduce accuracy, it facilitates the exploration of innovative methodologies, including the application of machine learning over traditional PRC.

Reviewer 3 Report

Dear authirs

Thanks for your comments.

It is ok

Author Response

Thank you for your continued feedback. From an English language perspective, we confirm that the English language of the manuscript text has been checked, and according to the reports, no issues were detected. In the newer parts of the manuscript, several English revisions have been made, which can be observed in the revised document. We sincerely appreciate your attention to detail and valuable insights.